# Preparation and Characterization of Electrospun Pectin-Based Films and Their Application in Sustainable Aroma Barrier Multilayer Packaging

**Busra Akinalan Balik [1], Sanem Argin [1]** 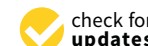**, Jose M. Lagaron [2,\*]** **and Sergio Torres-Giner [2,\*]**

[1] Department of Food Engineering, Yeditepe University, Atasehir, 34755 Istanbul, Turkey;
   busra.akinalan@yeditepe.edu.tr (B.A.B.); sanem.argin@yeditepe.edu.tr (S.A.)

[2] Novel Materials and Nanotechnology Group, Institute of Agrochemistry and Food Technology (IATA),
   Spanish National Research Council (CSIC), Calle Catedrático Agustín Escardino Benlloch 7, 46980 Paterna,
   Spain

\* Correspondence: lagaron@iata.csic.es (J.M.L.); storresginer@iata.csic.es (S.T.-G.);
   Tel.: +34-963-900-022 (J.M.L. & S.T.-G.)

**Featured Application:** The present study aims to develop novel pectin-based films by electrospinning. The here-prepared films were applied as aroma barrier interlayers between two biopolymer films to develop fully bio-based and biodegradable food packaging articles according to the principles of the Circular Economy.

**Abstract:** Pectin was first dissolved in distilled water and blended with low contents of polyethylene oxide 2000 (PEO$_{2000}$) as the carrier polymer to produce electrospun fibers. The electrospinning of the water solution of pectin at 9.5 wt% containing 0.5 wt% PEO$_{2000}$ was selected as it successfully resulted in continuous and non-defected ultrathin fibers with the highest pectin content. However, annealing of the resultant pectin-based fibers, tested at different conditions, developed films with low mechanical integrity, high porosity, and also dark color due to their poor thermal stability. Then, to improve the film-forming process of the electrospun mats, two plasticizers, namely glycerol and polyethylene glycol 900 (PEG$_{900}$), were added to the selected pectin solution in the 2–3 wt% range. The optimal annealing conditions were found at 150 °C with a pressure of 12 kN load for 1 min when applied to the electrospun pectin mats containing 5 wt% PEO$_{2000}$ and 30 wt% glycerol and washed previously with dichloromethane. This process led to completely homogenous films with low porosity and high transparency due to a phenomenon of fibers coalescence. Finally, the selected electrospun pectin-based film was applied as an interlayer between two external layers of poly(3-hydroxybutyrate-*co*-3-hydroxyvalerate) (PHBV) by the electrospinning coating technology and the whole structure was annealed to produce a fully bio-based and biodegradable multilayer film with enhanced barrier performance to water vapor and limonene.

**Keywords:** pectin; electrospinning; annealing; barrier interlayers; food packaging

## 1. Introduction

Pectin is one of the most abundant polysaccharides in nature, which is found in the middle lamella of cell wall, primary cell walls, and plasma membrane of plants [1,2]. It is commercially produced by the industrial waste of apple pomace, citrus peel, and sugar beet pulp [3]. Pectin is mainly composed of linear chains of α-1,4 linked ᴅ-galacturonic acid units but also different types of side chains may exist in its chemical structure such as those containing rhamnose, xylose, galactose, and arabinose [2,4]. Depending on the side chains, pectin domains are named differently such as homogalacturonan (HGA),

rhamnogalacturonan-I (RG-I), rhamnogalacturonan-II (RG-II), and xylogalacturonan (XGA) [5,6]. HGA units are habitually referred as 'smooth' regions of pectin and it comprises galacturonic acid groups whereas other units are called 'hairy' regions [4,7].

Pectin can be used as gelling, thickening, and stabilizing agent in the food and pharmaceutical industry. Moreover, the polyelectrolyte nature, biodegradability, biocompatibility, and water solubility of pectin also open up new uses including coatings or edible films for food packaging applications [8,9]. Nevertheless, the intrinsically high hydrophilicity and low mechanical strength of pectin-based films compared to conventional ones such as those made of high-density polyethylene (HDPE) and polypropylene (PP) currently limit the application of this carbohydrate [10–12]. In this context, to enhance the mechanical integrity and also reduce brittleness, plasticizers are habitually added to the pectin formulations to form the films [13]. The addition of different kinds of plasticizers, for instance glycerol [14–16], sorbitol [13,17,18], polyethylene glycol (PEG) [13,19] or xylitol [20], to polysaccharide film-forming solutions during the casting process can improve both film formation and flexibility by enhancing the motion of polymer chains and reducing their intra- and intermolecular forces [21,22]. Additionally, water dissolution of pectin can be decreased by in situ cross-linking with divalent metal ions including $Ca^{2+}$, $Zn^{2+}$ or $Mg^{2+}$ [23,24]. Furthermore, the mechanical, thermal, and barrier properties of the pectin-based films can also be enhanced by blending with other biopolymers such as chitosan [25,26], cellulose, and its derivatives [27,28] or the addition of inorganics such as nanoclays [11].

In the polymer literature, neat pectin films are technically not feasible to be produced by conventional melt technologies and they have been so far obtained by the solvent casting method using large contents of plasticizers [24,25,29,30]. In this regard, electrospinning is a novel technique that provides manufacturing of ultrathin fibers with diameters extending from several nanometers to a few micrometers [31]. Electrospun nanofibers may offer many functional advantages such as superior mechanical properties, large surface-to-mass ratio, tailored fiber morphology, and the capability of encapsulation and subsequent release of active and bioactive principles [32–36]. However, aqueous solutions of neat pectin cannot be electrospun due to the limited viscoelasticity of pectin and its insufficient chain entanglements [37–39]. As a result, electrospun pectin nanofibers have been only obtained by blending with different synthetic polymers such as polyethylene oxide (PEO) [39–45], polyvinyl alcohol (PVOH) [46–48], and pullulan [38]. In addition, ternary blends of alginate/pectin/PEO [49] and chitosan/pectin/PVOH [50] have been recently successfully electrospun for biomedical purposes.

The few studies reporting the development of electrospun pectin-based nanofibers have been mainly focused on the areas of antibacterial surfaces [48], tissue engineering [40–42,50], drug delivery [44], and encapsulation [38,49], whereas their utilization for food packaging applications remains unexplored due to the inherent discontinuity and porous structure of the nanofibers mats. Interestingly, electrospun mats can be subjected to a thermal post-treatment above the glass transition ($T_g$) and below the melting temperature ($T_m$) of the polymer, also termed annealing, in order to remove or minimize their porosity and produce continuous and homogenous films [51–55]. Until now, this technology has been successfully applied to different polyester-type biopolymers with different potential applications in the food packaging field. For instance, electrospun poly(3-hydroxybutyrate) (PHB) films showed better optical properties, similar barrier performance, and higher elongation at break and toughness in comparison with equivalent films obtained by compression molding [51]. Electrospun films of PHB, PVOH, and also polylactide (PLA) were also developed by electrospinning and originally applied as coating materials on a paper-based packaging material to develop multilayers with improved barrier properties against water and limonene vapors [52]. In another study carried out by Cherpinski et al. [53], a similar strategy was followed to coat cellulose nanopapers by PHB and poly(3-hydroxybutyrate-*co*-3-hydroxyvalerate) (PHBV) electrospun layers. Similarly, electrospun ultrathin fibers of bio-waste derived PHBV were subjected to annealing by Melendez-Rodriguez et al. [54] to successfully produce continuous biopolymer films with similar barrier performance than petroleum-based polyethylene terephthalate (PET) films. Other recent

studies have been focused on the incorporation of antimicrobial or antioxidant ingredients in the electrospun fibers, which can be thereafter integrated as active layers in packaging structures. For example, Figueroa-Lopez et al. [56] prepared electrospun active films of PHBV with antimicrobial and antioxidant properties by the incorporation into the fibers of different essential oils (EOs) and natural extracts (NEs). Also, Quiles-Carrillo et al. [57] recently developed multilayer bioactive films with controlled release capacity of the natural antioxidant gallic acid (GA) by the incorporation of electrospun PLA interlayers into cast-extruded PLA films. Lastly, Radusin et al. [58] recently prepared antimicrobial PLA films containing *Allium ursinum* L. extract by electrospinning.

The aim of this study was to obtain, for the first time, electrospun pectin-based films as potential candidates for food packaging applications. First, various water-based pectin solutions containing different amounts of PEO and/or in combination with the addition of two different types of plasticizers were processed by electrospinning to determine the best system to produce a film. Thereafter, the selected electrospun mats were characterized and the most promising fibers were selected and subjected to annealing to produce pectin-based films. The morphology, chemical, and thermal properties of the films were reported. The optimal electrospun film was, finally, applied as an interlayer in a multilayer structure based on PHBV and the barrier properties of the resultant multilayers were analyzed and compared to an equivalent multilayer of a cast-film pectin interlayer.

## 2. Materials and Methods

### 2.1. Materials

Low methyl esterified amidated pectin was kindly received from AROMSA Inc. (Gebze, Turkey). The product (GENU pectin, LM-104 AS-FS, degree of esterification 27%, degree of amidation 20%) was produced and delivered in powder form by CP Kelco (Copenhagen, Denmark). PEO with molecular weight ($M_w$) of 2000 kDa, that is, $PEO_{2000}$, was obtained in powder form as SENTRYTM POLYOXTM WSR N80-LEO NF grade by The Dow Chemical Company (Midland, MI, USA). PEG with $M_w$ of 900 kDa, that is, $PEG_{900}$, was provided by Honeywell Fluka Chemicals Company (Bucharest, Romania). Bacterial aliphatic copolyester PHBV was ENMAT™ Y1000P, produced by Tianan Biologic Materials (Ningbo, China) and distributed by NaturePlast (Ifs, France). The product was delivered as off-white pellets packaged in plastics bags. The biopolymer resin presents a true density of 1.23 g/cm$^3$ and the pellets a bulk density of 0.74 g/cm$^3$, as determined by ISO 1183 and ISO 60, respectively. Sorbitan monolaurate was obtained from Sigma-Aldrich S.A. (Madrid, Spain) as Span$^®$ 20. According to the manufacturer, its fatty acid composition was lauric acid (C12:0) ≥ 44%; balance primarily myristic (C14:0), palmitic (C16:0), and linolenic (C18:3) acids. Calcium chloride, dichloromethane, 2,2,2-trifluoroethanol (TFE), ≥99% purity, and glycerol, ≥99.5% purity, were all also purchased from Sigma-Aldrich S.A. (Madrid, Spain).

### 2.2. Preparation of Solutions

The total concentration of solids in distilled water to prepare the fiber-forming solution for electrospinning was set at 10 wt%. For this, pectin was first dissolved in water at 70 °C for 3 h and the solution was gently stirred overnight at room temperature. Then, $PEO_{2000}$ with or without a plasticizer, that is, glycerol or $PEG_{900}$, was added to the pectin solution and it was further stirred for 24 h. In all cases, Span$^®$ 20 was added as a surfactant to the pectin solutions at 2 wt% with respect to the total solid weight content of the solution. Table 1 summarizes the compositions of the water-based solutions prepared for electrospinning. For the electrospinning of the PHBV layers, the copolyester resin was dissolved at 10% (w/v) in TFE at room conditions during 24 h.

**Table 1.** Different solutions prepared according to the weight content (wt%) of pectin, polyethylene oxide 2000 ($PEO_{2000}$), glycerol, and polyethylene glycol 900 ($PEG_{900}$) in distilled water.

| Solution | Pectin (wt%) | $PEO_{2000}$ (wt%) | Glycerol (wt%) | $PEG_{900}$ (wt%) | Water (wt%) |
|----------|--------------|----------------------|----------------|---------------------|-------------|
| S1 | 9.9 | 0.1 | - | - | 90 |
| S2 | 9.75 | 0.25 | - | - | 90 |
| S3 | 9.5 | 0.5 | - | - | 90 |
| S4 | 9.0 | 1.0 | - | - | 90 |
| S5 | 7.5 | 0.5 | 2.0 | - | 90 |
| S6 | 7.0 | 0.5 | 2.5 | - | 90 |
| S7 | 6.5 | 0.5 | 3.0 | - | 90 |
| S8 | 7.5 | 0.5 | - | 2.0 | 90 |
| S9 | 7.0 | 0.5 | - | 2.5 | 90 |
| S10 | 6.5 | 0.5 | - | 3.0 | 90 |

## 2.3. Characterization of Solutions

Prior to electrospinning, all the prepared pectin solutions were characterized in terms of their viscosity, surface tension, and conductivity. Solution viscosity was determined by a VISCO BASIC Plus L rotational viscosity meter equipped with a low-viscosity adapter (LCP) from Fungilab S.A. (San Feliu de Llobregat, Spain). Conductivity was measured in a conductivity meter XS Con6 from Lab-box (Barcelona, Spain). Surface tension was determined in an EasyDyne tensiometer K20 model Krüss GmbH (Hamburg, Germany) following the Wilhelmy plate method. All measurements were carried out at room temperature in triplicate.

## 2.4. Electrospinning

An electrospinning machine Fluidnatek® LE500 (Bioinicia S.L., Valencia, Spain) placed in a closed chamber and connected to an environmental control unit was used in the lab mode. Fibers were collected in vertical mode on a rectangular plate collector covered with aluminum foil. The most suitable conditions for the electrospinning of pectin were determined during the process. The flow-rate of the pectin-base solutions was set at the highest value possible in order to attain the maximum yield. Flow-rate was optimal at 3 mL/h since higher values led to some droplets on the collector. The tip-to-collector distance was also adjusted by decreasing it until the fibers were formed, being optimal at 25 cm. Finally, the applied voltage was smoothly increased up to the point a stable jet was obtained. The most optimal values of voltage ranged from 16 to 20 kV. For the electrospinning of PHBV, the voltage was set at 10 kV, the tip-to-collector distance was 15 cm, and the flow-rate was 6 mL/h. These values were selected based on our previous study [54]. All the experiments were conducted at 25 °C and 30% relative humidity (RH).

## 2.5. Washing and Drying

The resultant electrospun pectin-based fibers were washed by soaking the mats into dichloromethane for 60 s. Dichloromethane was chosen since pectin is not soluble in this solvent [39], but it could facilitate fiber coalescence by reducing the porosity of the electrospun mats. The washed mats were then placed in a Vaciotem-TV (P. Selecta, Barcelona, Spain) vacuum drying oven connected to a Vacuubrand vacuum pump at 27 °C and 100 mmHg pressure for 18 h in order to remove the organic solvent.

## 2.6. Films Preparation

The washed and dried fibers were then subjected to annealing in a 4122-model press from Carver, Inc. (Wabash, IN, USA). A set of experiments were conducted to select the optimal temperature, time, and load to produce homogenous and transparent films. To this end, the electrospun mats were post-treated in the temperature range of 50–240 °C and the pressure range of 6–30 kN for times ranging

from 5 s to 120 s. A pectin film was also prepared by casting as a control material. To this end, 2 g of pectin powder was dissolved in 100 mL of distilled water and then 0.92 g of glycerol was added. After 24 h of mixing, 10 mL of solution was poured into polystyrene (PS) petri dishes and left at room conditions, that is, 25 °C and 40%, for 3 days.

The multilayer films were prepared by placing either the solvent-casted or electrospun pectin-based films as an interlayer between two electrospun layers of PHBV. This was accomplished by electrospinning PHBV fibers on one side of the previously prepared pectin films. The resultant coated films were turned down and coated on the other side. The two side coated films were then placed in the press and annealed at 160 °C for 10 s, without pressure, based on our previous research [55]. These annealing conditions were selected since the film-forming process is controlled by the external layers, which are habitually thicker. Control films made of two electrospun layers of PHBV without pectin were also prepared in the same conditions.

### 2.7. Characterization of the Electrospun Fibers and Films

#### 2.7.1. Thickness and Conditioning

Prior to testing, the thickness of the electrospun mats and films was measured using a digital micrometer series S00014 (Mitutoyo Corporation, Kawasaki, Japan) with ±0.001 mm accuracy. Measurements were performed at five random positions and values were averaged. The samples were stored in a desiccator at 25 °C and 0% RH for 24 h before characterization.

#### 2.7.2. Morphology

The morphologies of the electrospun fibers and the top views and cross-sections of the pectin-based films were investigated by a scanning electron microscope (SEM, Hitachi S-4800, Tokyo, Japan). The samples were cryo-fractures using nitrogen liquid. Prior to analysis, all the samples were coated with a gold/palladium alloy for 2 min by a Polaron sputter coater (Quarum Technologies, Kent, UK). A 5 kV voltage was applied during SEM analysis. Fiber diameters and layer thicknesses were determined by the software ImageJ, Java v.1.52a from the measurement of, at least, 50 fibers.

#### 2.7.3. Thermal Analysis

The thermal properties of $PEO_{2000}$ and pectin powders as well as the electrospun pectin-based fibers obtained from S3 and from S5 to S10 were determined by differential scanning calorimetry (DSC) and Thermogravimetric analysis (TGA). Thermal transitions were determined in a DSC-7 analyzer from PerkinElmer, Inc. (Waltham, MA, USA), equipped with a cooling accessory Intracooler 2 from PerkinElmer, Inc. Approximately 3 mg of sample was placed into the aluminum pan, while an empty pan was used as reference. Calibration was previously conducted using an indium sample. The samples were first heated from −70 °C to 160 °C, then cooled back to −70 °C, and then heated again to 300 °C. The heating and cooling rates were set at 10 °C/min. The experiments were conducted under nitrogen atmosphere and all DSC tests were performed in triplicate.

To ascertain their thermal stability, TGA was performed under nitrogen atmosphere in a Thermobalance TG-STDA Mettler Toledo model TGA/STDA851e/LF/1600 analyzer (Greifensee, Switzerland). TGA curves were obtained after conditioning the samples in the sensor for 5 min at 30 °C. The samples were heated from 25 °C to 700 °C at a heating rate of 10 °C/min. All TGA tests were also carried out in triplicate.

#### 2.7.4. Fourier Transform Infrared Spectroscopy

Fourier transform infrared spectroscopy (FTIR) spectra of the raw materials, that is, the pectin powder, glycerol, and $PEO_{2000}$, and also the electrospun fibers and film obtained from S7 solution were obtained from an average of 20 scans by a Bruker Tensor 37 (Rheinstetten, Germany) spectrometer connected with a Golden Gate of Specac, Ltd. (Orpington, UK) attenuated total reflection (ATR)

accessory. ATR-FTIR was performed in order to analyze the existence of any chemical interactions between the components. The scans were collected in the wavelength values from 4000 cm$^{-1}$ to 600 cm$^{-1}$ at a resolution of 4 cm$^{-1}$.

### 2.7.5. Color Measurements

The color of the pectin-based interlayers and the resultant multilayers in PHBV was carried out in a benchtop spectrophotometer Konica Minolta CM-5, from Hunter Associates Laboratory, Inc. (Reston, VA, USA). The Commission Internationale de l'Eclairage (CIE) standard illuminant D65 was used to assess the CIE Lab color space coordinates L*a*b* using an observer angle of 10°. L* represents the luminance (black to white), a* indicates the change between green and red, and b* represents the change from blue to yellow. The colorimeter was calibrated with a white standard tile and a mirror device for the black (no light reflection). The L*a*b* coordinate values were obtained on five different samples and the color difference (ΔE*) was calculated following Equation (1).

$$\Delta E^{*} = \left[ (\Delta L^{*})^2 + (\Delta a^{*})^2 + (\Delta b^{*})^2 \right]^{0.5} \tag{1}$$

where ΔE*, Δa*, and Δb* corresponded to the differences between the color parameters of the multilayer samples and the values of the PHBV/PHBV multilayer. Color change was evaluated as follows: Unnoticeable ($\Delta E^{*}_{ab} < 1$), only an experienced observer can notice the difference ($\Delta E^{*}_{ab} \geq 1$ and <2), an unexperienced observer notices the difference ($\Delta E^{*}_{ab} \geq 2$ and < 3.5), clear noticeable difference ($\Delta E^{*}_{ab} \geq 3.5$ and < 5), and the observer notices different colors ($\Delta E^{*}_{ab} \geq 5$) [59].

### 2.7.6. Permeability Tests

#### Water Vapor Permeance

The water vapor permeance of the multilayer films was measured according to the ASTM 2011 gravimetric method. In order to conduct this test, 5 mL of distilled water was put inside a Payne permeability cup (Inner diameter = 3.5 cm) (Elcometer Sprl, Belgium). The films were located in the cups so that on one side they were exposed to 100% RH, avoiding direct contact with water. Then, the cups were locked with silicon rings and kept in a conditioned desiccator (25 °C and 0% RH). The control samples were cups with aluminum films to estimate solvent loss through the sealing. The cups were weighed regularly for every 24 h using an analytical balance having an accuracy of ±0.0001 g, until the values reached a plateau. Water vapor permeation rate corresponded to the slope value of the steady state line of time versus weight loss per unit area and the weight loss was calculated as the total loss minus the loss through the sealing. Water permeance was obtained by correcting the water vapor permeation rate for the permeant partial pressure. Tests were conducted in triplicate.

#### Limonene Vapor Permeance

Limonene vapor permeance of the multilayer films was determined as similar as described above for water vapor. For this 5 mL of ᴅ-limonene was placed inside the Payne permeability cups and the cups containing the films were placed at controlled room conditions of 25 °C and 40% RH. The limonene vapor permeation rates were estimated from the steady-state permeation slopes and the weight loss was calculated as the total cell loss minus the loss through the sealing. Limonene permeance was obtained by correcting the limonene vapor permeation rate for the permeant partial pressure. Tests were conducted in triplicate.

### 2.8. Statistical Analysis

All data were analyzed statistically by SPSS Statistics 17.0 (IBM, Chicago, IL, USA). Tukey's HSD test was used to determine the significant differences among samples ($p < 0.05$). Different

superscripts show statistically different results. Unlike groups of letters were used to express each group of properties.

## 3. Results and Discussion

### 3.1. Preparation of Electrospun Pectin-Based Fibers

The morphology of the resultant electrospun fibers is shown in Figure 1. A solution of pure pectin was initially also tested but, instead of continuous jets, it formed large droplets when subjected to high voltages during electrospinning due to the limited viscoelasticity and insufficient chain entanglements of the carbohydrate [37–39]. A similar phenomenon was faced by Liu et al. [38] during the electrospinning of pectin. In order to increase the viscoelasticity of pectin, different quantities of $PEO_{2000}$ from 0.1 to 1.0 wt% were added to the pectin solutions for electrospinning, the here so-called S1 to S4. The addition of PEO can help reducing the repulsive forces among negatively charged pectin chains, enhancing chain entanglement and fiber formation [45]. The primary intention was to keep the $PEO_{2000}$ content at a minimum value in order to produce electrospun fibers with the highest content of pectin. In Figure 1a it can be observed that the electrospinning of the pectin solution having 0.1 wt% $PEO_{2000}$, that is, S1, resulted in fibers with a discontinuous and beaded morphology. The fibers produced from S2, shown in Figure 1b, which is based on 0.25 wt% $PEO_{2000}$, were more continuous and non-beaded. However, the resultant electrospun mat showed poor integrity as these easily fractured during detachment from the collector. In the case of S3, made of a pectin solution with 0.5 wt% $PEO_{2000}$, neat fibers free of beads and with a uniform diameter were produced as it can be observed in Figure 1c. Furthermore, branched and thick pectin fibers were obtained for the electrospinning of S4, which contained 1 wt% of $PEO_{2000}$ and are shown in Figure 1d. These results confirm that the use of a carrier polymer in an appropriate amount is a key parameter in order to achieve continuous and non-defected fibers during electrospinning. Based on these results, S3 was selected due to it contained the optimal amount of $PEO_{2000}$ that yielded the fibers with the highest uniformity and a relatively low diameter, whereas S1, S2, and S4 were discarded from the study. Although there was not any particular requirement for the diameter, it was necessary to obtain continuous and free-bead fibers in order to produce films. Thereafter, different types and amount of plasticizers were also tested to improve the film-forming capacity of the electrospun pectin mats. Therefore, glycerol or $PEG_{900}$ were added to the pectin solution based on the S3 composition in which the $PEO_{2000}$ content was kept constant at 0.5 wt%. One can observe in Figure 1e–j that similar morphologies, based on smooth and continuous fibers but slightly thinner in the case of $PEG_{900}$-containing fibers, were generated when the plasticizers were added. The average diameters of the electrospun pectin-based fibers are summarized in Table 2.

**Table 2.** Properties of the pectin-based solutions and mean diameters of their resultant electrospun fibers.

| Solution | Viscosity (cP) | Surface Tension (mN/m) | Conductivity (μS·cm) | Fiber Diameter (nm) |
|---|---|---|---|---|
| S1 | 1892 ± 78 [a] | 28.8 ± 0.2 [a] | 7.25 ± 0.12 [a] | - |
| S2 | 3066 ± 48 [b] | 28.4 ± 0.3 [a] | 6.37 ± 0.15 [a,b] | 156 ± 33 [a] |
| S3 | 5950 ± 219 [c,e] | 30.3 ± 0.2 [b,d] | 6.10 ± 0.10 [c] | 186 ± 32 [b] |
| S4 | 12,155 ± 1660 [d] | 32.9 ± 0.2 [c] | 5.99 ± 0.25 [b,c] | 299 ± 41 [c] |
| S5 | 5511 ± 299 [c] | 31.6 ± 0.2 [b,e,f] | 5.64 ± 0.37 [c,d] | 272 ± 43 [d] |
| S6 | 5751 ± 401 [c] | 30.9 ± 0.3 [d,e] | 5.30 ± 0.13 [c,d] | 304 ± 56 [c,d] |
| S7 | 6440 ± 327 [c,e] | 32.1 ± 0.2 [g,f] | 4.98 ± 0.09 [c,d] | 329 ± 42 [c,d] |
| S8 | 6047 ± 219 [e] | 31.2 ± 0.3 [e] | 5.11 ± 0.20 [c,e] | 189 ± 40 [b] |
| S9 | 6106 ± 124 [e] | 31.1 ± 0.3 [e] | 4.82 ± 0.21 [e] | 197 ± 44 [b] |
| S10 | 6317 ± 364 [e] | 32.5 ± 0.2 [g,c] | 4.72 ± 0.22 [e] | 223 ± 33 [b] |

[a–g] Different letters in the same column indicate a significant difference among the samples ($p < 0.05$).

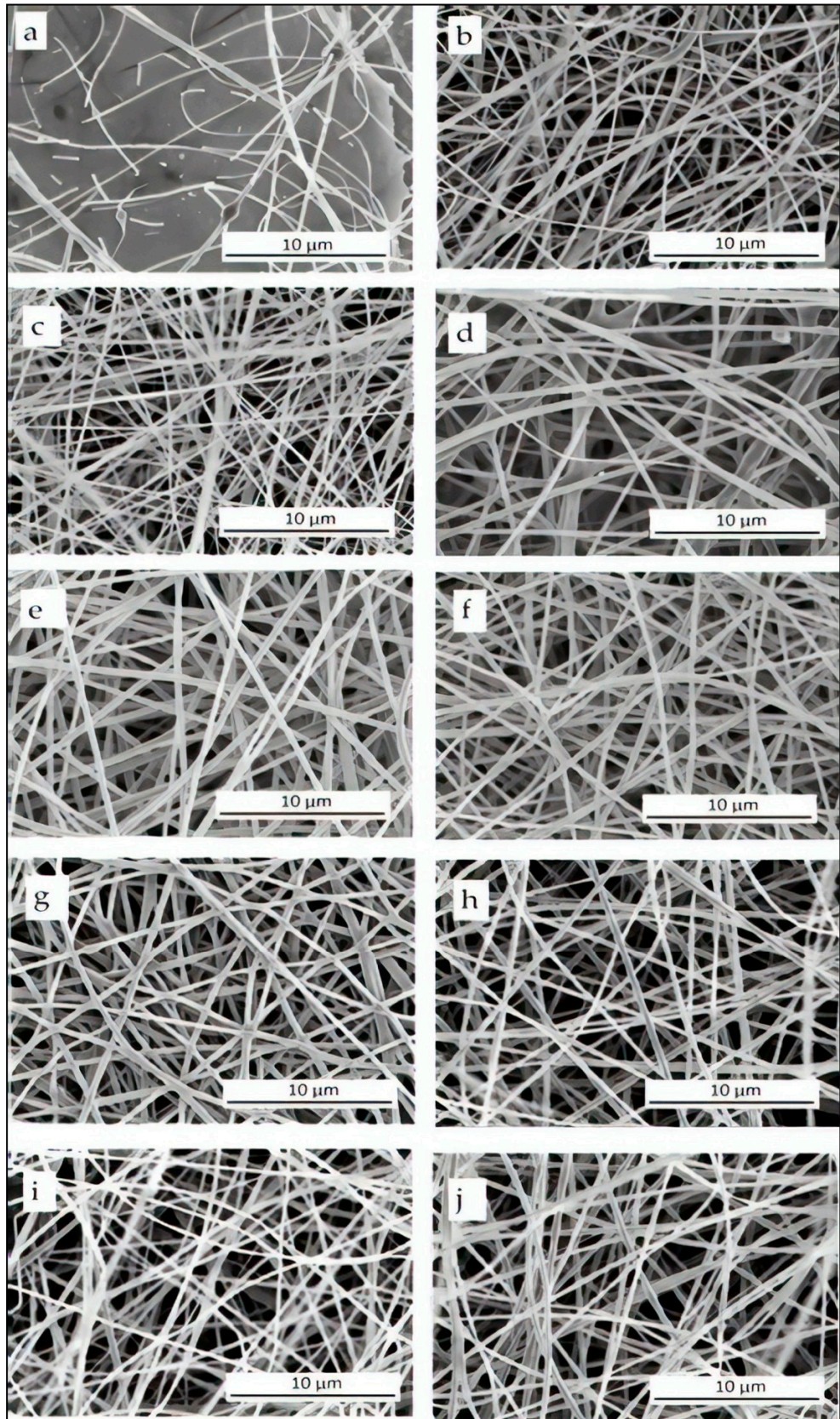

**Figure 1.** Scanning electron microscopy (SEM) images of the electrospun mats obtained from the pectin-based solutions of: (**a**) S1; (**b**) S2; (**c**) S3; (**d**) S4; (**e**) S5; (**f**) S6; (**g**) S7; (**h**) S8; (**i**). S9; (**j**) S10.

The solution properties were determined to better understand the morphologies of the ultrathin pectin-based fibers in the attained electrospun mats. The values of viscosity, surface tension, and conductivity of the fiber-forming solutions are also given in Table 2. For the non-plasticized solutions, that is, S1–S4, viscosity of the pectin-based solutions increased significantly when the amount of $PEO_{2000}$ was increased. One can observe that the addition of plasticizers did not create any significant difference in the solution viscosity. Even when the content of glycerol or $PEG_{900}$ was increased to 3 wt%, the solution viscosity did not change considerably, showing a value around 6000–6400 cP. The surface tension slightly increased from approximately 29 mN/m to values in the range of 31–33 mN/m when the $PEO_{2000}$ content was increased above 0.5 wt%. It was not observed a clear correlation between the plasticizer content and the surface tension of pectin-based solutions. It can also be observed that, due to the inherent polyelectrolyte nature of pectin [60], its aqueous solutions were highly conductive, showing values in the 6–8 μS.cm range. However, the conductivity values decreased up to values close to 4 μS.cm when the $PEO_{2000}$ content was increased and, particularly, when the plasticizers were added since the amount of pectin in the solution was reduced. Therefore, the present results suggest that the fiber formation was attained due to a combined effect of viscosity increase and conductivity decrease. The optimal values were particularly attained in the range of ~5500–6500 cP and ~5–6 μS.cm of viscosity and conductivity, respectively. Therefore, moderate-to-high viscosities in combination with relatively low conductivities tended to produce the fibers with the most optimal morphology, while the effect of surface tension was negligible. As previously indicated in other works describing the role of solution properties in the electrospinnability of biopolymers [32], it was difficult to elucidate the effect of a single property without considering the impact of the other ones.

In relation to the plasticizers, one can observe that the diameters of the fibers varied from 156 ± 33 nm to 329 ± 42 nm, which could be related to differences in the solution properties described above. Significantly thicker fibers were attained with the increase of the $PEO_{2000}$ content (from S2 to S4) for the non-plasticized samples since the viscosity of these solutions was increased with the $PEO_{2000}$ content [49]. Also, the incorporation of glycerol to the $PEO_{2000}$ containing fibers resulted in significantly thicker fibers. This increase in the fiber diameter can be related to the increase in the solution viscosity. As shown in Table 2, S3 showed a viscosity of 5950 cP and it yielded fibers with average diameter of 186 ± 32 nm, whereas fibers with an average diameter of 329 ± 42 nm was attained in S7 with a viscosity of 6440 cP. This observation can be ascribed to the plasticizing effect of glycerol, which favored the molecular entanglements of the pectin chains by reducing their intermolecular interaction. The latter effect is based on the fact that intermolecular H-bonds of pectin can be substituted by glycerol-pectin H-bonds and covalent esters, thus the attraction between pectin molecules is reduced [29]. A similar phenomenon was also reported by Cui et al. [40] where glycerol-containing solutions resulted in pectin-based fibers with higher diameters than those obtained from solutions using dimethylformamide (DMF) or dimethyl sulfoxide (DMSO) as co-solvents. Also, if one compares the solutions containing 2.5 wt% of plasticizer, which were labelled as S5 and S8, the glycerol-containing ones yielded fibers with an average diameter of 272 ± 43 nm, while the solutions with $PEG_{900}$ produced fibers in the range of 189 ± 40 nm. This can be explained by the fact that $PEG_{900}$ has a higher $M_w$ than glycerol and, thus, less substitution of H-bonds could have been occurred and therefore less chain entanglements were formed. As mentioned earlier, electrospinning of the 1 wt% PEO containing solution, that is, S4, resulted in pectin-based fibers with a discontinuous and beaded morphology, which is a consequence of the formation of a solution with low viscosity and high conductivity.

### 3.2. Thermal Properties of Electrospun Pectin-Based Fibers

The DSC curves corresponding to the cooling and second heating steps of the as-received $PEO_{2000}$ and pectin powders and the electrospun pectin-based fibers obtained from S3 and S5–S10 are gathered in Figure 2. During the cooling process, shown in Figure 2a, one can observe that $PEO_{2000}$ crystallized from the melt showing a crystallization temperature ($T_c$) of approximately 40 °C [61]. Alternatively, the pectin powder showed no crystallization during cooling in the whole tested thermal range. The

crystallization peak attributed to PEO$_{2000}$ was not observed in the electrospun PEO$_{2000}$-containing pectin fibers due to its relative low content. However, a small exothermic peak was observed at approximately 23 °C for the electrospun pectin fibers obtained from S10, which can be attributed to the crystallization of the PEG$_{900}$ confined in the carbohydrate [61,62]. In Figure 2b one can observe the melting temperature (T$_m$) of PEO$_{2000}$ at nearly 68 °C and also two low-intense melting peaks corresponding to the melting of the PEG$_{900}$ fraction in the pectin fibers obtained from S10 at approximately 24 °C and 40 °C [62]. Pectin did not exhibit any thermal transition of first order, that is, crystallization or melting, thus confirming the carbohydrate is fully amorphous [13,63]. Moreover, no glass transition was observed up to nearly 230 °C, temperature from which the carbohydrate started thermal degradation. Previous studies on the thermal properties of pectin reported that thermal degradation of this carbohydrate is an exothermic process [63,64]. It is worthy to note that the degradation temperature (T$_{deg}$) of pectin shifted from 232 °C, for the neat powder, to 236 °C, for the electrospun fibers obtained from S10, which suggests that the thermal stability of the carbohydrate was slightly enhanced by the addition of PEG$_{900}$. This improvement can be related to the newly formed intermolecular PEG-pectin H-bonds and covalent esters, which delayed the thermal degradation of the pectin macromolecule [29].

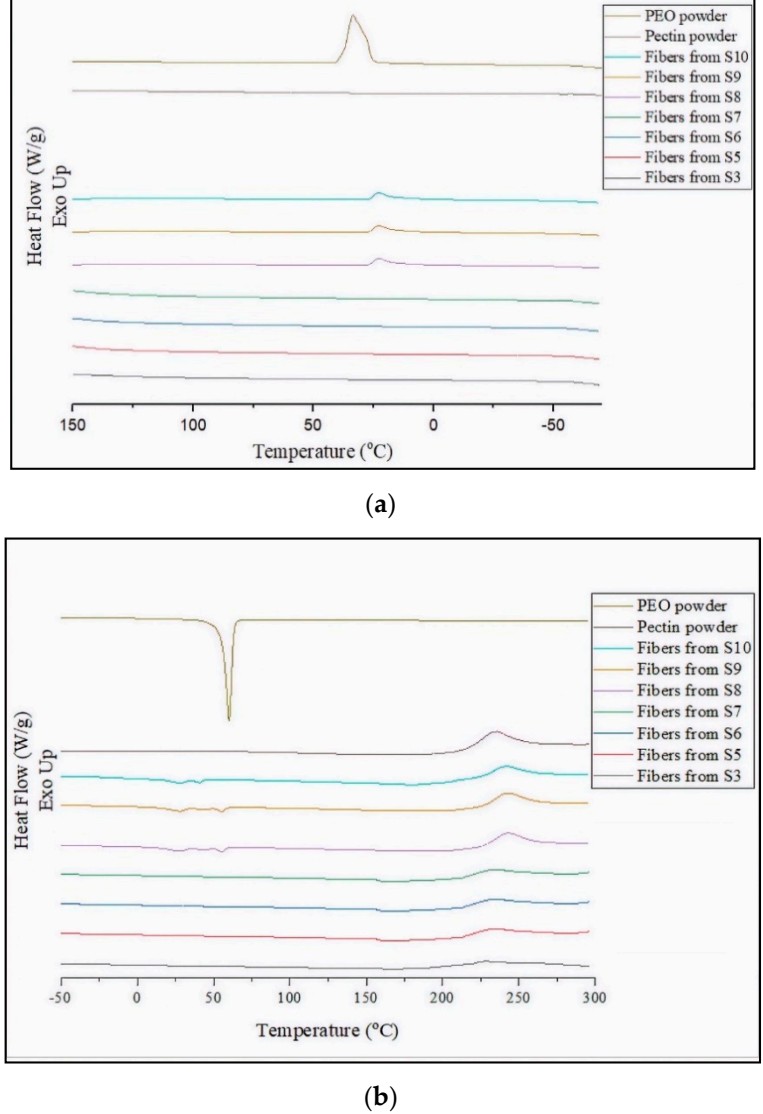

**(a)**

**(b)**

**Figure 2.** Differential scanning calorimetry (DSC) curves of polyethylene oxide 2000 (PEO$_{2000}$) powder, pectin powder, and electrospun pectin-based fibers during: (**a**) cooling and (**b**) second heating.

The TGA curves of the neat pectin and PEO$_{2000}$ powders as well as the electrospun pectin-based fibers obtained from S3 and S5–S10 are shown in Figure 3. The most relevant thermal parameters obtained from the TGA curves are given in Table 3. One can notice that pectin exhibited three significant main weight losses, occurring at approximately 100 °C, 217 °C, and 240 °C. The first mass loss, which took place in the 75–110 °C range, can be ascribed to the removal of bound water from the carbohydrate due to its highly hydrophilic nature. In this regard, Gloyna et al. [65] indicated that water evaporation of pectin observed between 50 °C and 150 °C, with a maximum at 100 °C, is identical to the dehydration process of other polysaccharides. Kastner et al. [66] and Nisar et al. [67] also observed a second weight loss between 195–350 °C. According to their studies, pectin shows a two-step degradation process in this range that might be related to the cracking of bonds or functional groups, structural depolymerization, and chain breaking of the polysaccharide. The carbohydrate then lost 60% of its total weight up to 350 °C whereas, in the 350–697 °C range, the mass loss was 15% of its original weight and the maximum rate of weight loss was observed at 235 °C. During thermal degradation of pectin different depolymerization reactions occur, including demethoxylation, depolymerization by backbone hydrolysis and hydrolytic cleavage of neutral sugar side chains [66–68]. The most instable bonds, formed by the neutral sugars, thermally degrade first. The polygalacturonic acid units of high $M_W$ hydrolyze later whereas the glycoside bonds formed by uronic acids also degrade at higher temperatures. Depending on the pectin properties, that is, pH, source, degree and pattern of methyl esterification, acid hydrolysis or β-elimination reactions take place during the thermal degradation [69]. In particular, the here-used low methyl esterified amidated pectin results in acid hydrolysis reactions with the temperature increase [70,71]. During hydrolysis, longer chains break up to the shorter as the cleavage of α-(1,4)-glycosidic bond connecting two uronic acids by the addition of a water molecule [72]. In our study, above 250 °C, a secondary degradation of pectin has been reported to occur, including release of functional side groups and chains break, while gasification of char residues arises at temperatures around 600 °C [73].

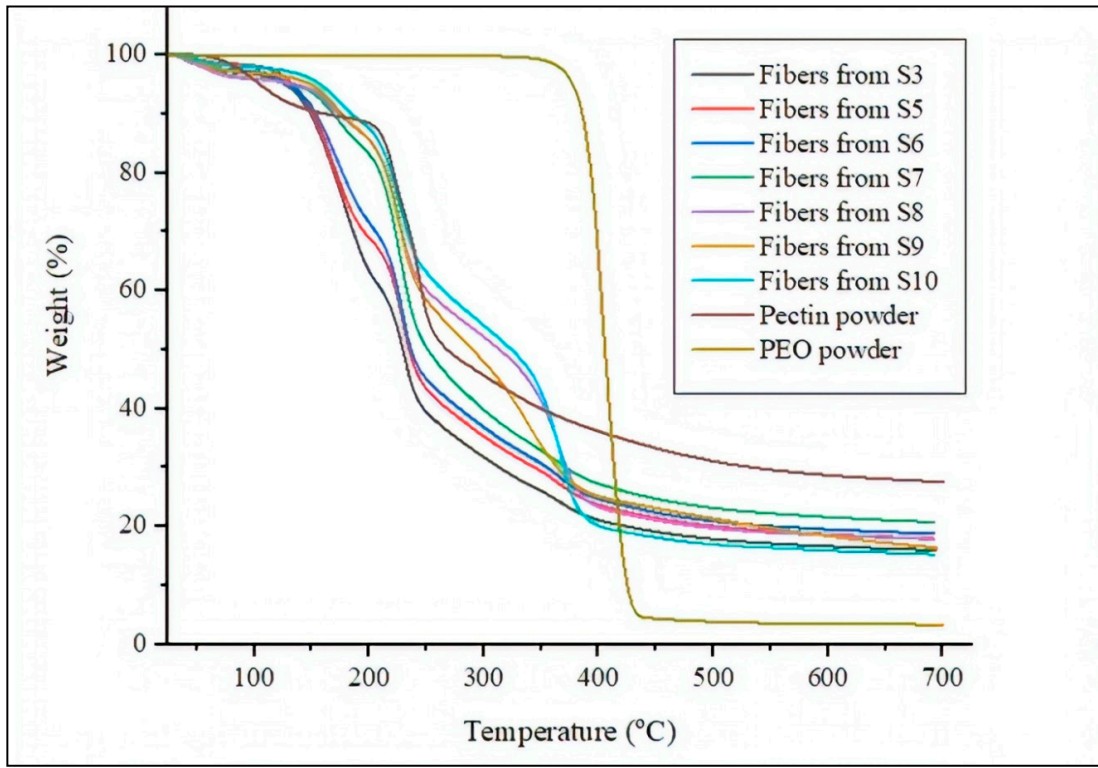

**Figure 3.** Thermogravimetric analysis (TGA) curves of the neat pectin powder, PEO$_{2000}$ powder, and electrospun pectin-based fibers.

**Table 3.** Main thermal parameters of the neat pectin powder, $PEO_{2000}$ powder, and the electrospun pectin-based fibers in terms of: onset temperature of degradation ($T_{onset}$), degradation temperature ($T_{deg}$), and residual mass at 700 °C.

| Sample | $T_{onset}$ (°C) | $T_{deg1}$ (°C) | $T_{deg2}$ (°C) | $T_{deg3}$ (°C) | Residual Mass (%) |
|---|---|---|---|---|---|
| Pectin powder | 98.0 ± 5.2 [a] | 217.4 ± 3.2 [a] | 240.2 ± 2.1 [a] | - | 27.4 ± 3.2 [a] |
| $PEO_{2000}$ powder | 375.3 ± 2.4 [b] | - | - | 400.0 ± 2.1 [a] | 23.7 ± 1.2 [b] |
| Fibers from S3 | 175.4 ± 2.5 [c] | 230.5 ± 3.2 [b] | - | 367.5 ± 3.2 [b] | 19.2 ± 1.1 [c] |
| Fibers from S5 | 169.3 ± 3.7 [c,d] | 230.3 ± 3.1 [b] | 297.3 ± 1.1 [b] | 367.3 ± 2.1 [b] | 20.2 ± 1.9 [c] |
| Fibers from S6 | 169.5 ± 2.1 [c,d] | 229.8 ± 2.0 [b] | 297.5 ± 1.2 [b] | 367.1 ± 2.0 [b] | 21.2 ± 2.0 [b,c] |
| Fibers from S7 | 170.3 ± 4.1 [c,d] | 227.5 ± 2.1 [b,c] | 297.4 ± 1.1 [b] | 367.3 ± 3.2 [b] | 23.4 ± 1.5 [b] |
| Fibers from S8 | 165.5 ± 5.2 [d] | 226.4 ± 2.2 [c] | - | 366.4 ± 2.6 [b] | 21.2 ± 1.3 [b,c] |
| Fibers from S9 | 168.2 ± 4.4 [d] | 226.2 ± 2.1 [c] | - | 367.4 ± 3.1 [b] | 20.0 ± 1.6 [c] |
| Fibers from S10 | 173.6 ± 5.0 [c,d] | 227.2 ± 3.0 [b,c] | - | 369.5 ± 2.0 [b] | 17.3 ± 1.2 [d] |

[a–d] Different letters in the same column indicate a significant difference among the samples ($p < 0.05$).

Alternatively, it can be observed that $PEO_{2000}$ was highly thermally stable, showing the values of onset degradation temperature ($T_{onset}$) and maximum degradation temperature ($T_{deg}$) of approximately 375 °C and 400 °C, respectively. One can also observe that the incorporation of $PEO_{2000}$ successfully delayed the thermal degradation of pectin up to nearly 175 °C. The reason of this behavior can be explained by the occurrence of more chain entanglements and the newly formed intermolecular PEO-pectin H-bonds and covalent esters, leading to the formation of a more stable and less volatile material. The pectin-based fiber obtained from the electrospinning of S3, S5, and S6 showed a similar thermal degradation profile than the neat pectin powder but with even lower values of $T_{deg}$. This result suggests that, even though $PEO_{2000}$ delays the onset of degradation of pectin, it also catalyzes its thermal degradation. Interestingly, the presence of both plasticizers contributed to increasing the thermal degradation of the $PEO_{2000}$-containing pectin fibers. The pectin-based fibers obtained from the electrospinning of S5–S10, which included glycerol or $PEG_{900}$, showed a similar thermal degradation profile in comparison with the $PEO_{2000}$-containing pectin fibers but with slightly lower values of $T_{onset}$. Additionally, the new weight loss obtained nearly 290 °C can be attributed to glycerol degradation. One can then consider that the plasticizers played a role in the pectin chains motion that favored chain scission during thermal degradation. Similar results were found for chia mucilage-glycerol films in the study performed by Dick et al. [74], showing that the glycerol addition lowered the heat resistance of the carbohydrate. Moreover, lower residual masses were observed for the plasticizer-containing fibers compared with the non-plasticized fibers. This effect was particularly intense for the electrospun pectin fibers obtained from S10, that is, the solution containing 3 wt% $PEG_{900}$, which showed a residual mass of 17 wt%.

### 3.3. Film-Forming Process of Electrospun Pectin-Based Fibers

The electrospun mats of pectin fibers were subjected to annealing in order to eliminate or minimize the porosity and then produce homogenous and continuous films. Based on the DSC results shown above, pure pectin is fully amorphous. Therefore, a heat treatment in a wide range of temperatures, that is, 50–220 °C, was needed to be applied for different processing times, that is, from 5s to 120 s, to find out the best conditions. Figure 4 shows the visual aspect of the different electrospun films obtained after annealing and Figure 5 includes the SEM micrographs of their surface fracture. However, one can observe that, at any of these conditions, the annealing failed to provide homogenous films using the fibers obtained from the non-plasticized pectin solution, that is, from S3. Moreover, the color of films became darker when subjected to temperatures above 160 °C due to the low onset temperature of degradation of pectin fibers, which is shown above in Table 3. As seen in Figure 4a,b, by increasing the temperature or time, not only the pectin materials developed a dark color but also the fiber mats lost their integrity. Therefore, based on these results, one can consider that the non-plasticized pectin-based

fibers were too rigid to gain flexibility by the only application of heat. Therefore, pressure was also applied during annealing to promote fibers rearrangement and the removal of porosity. In Figure 4c it can be observed that some transparent regions along the pectin-based materials were successfully attained when the mats were subjected to temperatures above 165 °C with an applied pressure of 12–24 kN for 1 min. Nevertheless, the resultant materials also exhibited an intense brownish color, partially lost their integrity and also became too brittle to be applied as packaging materials.

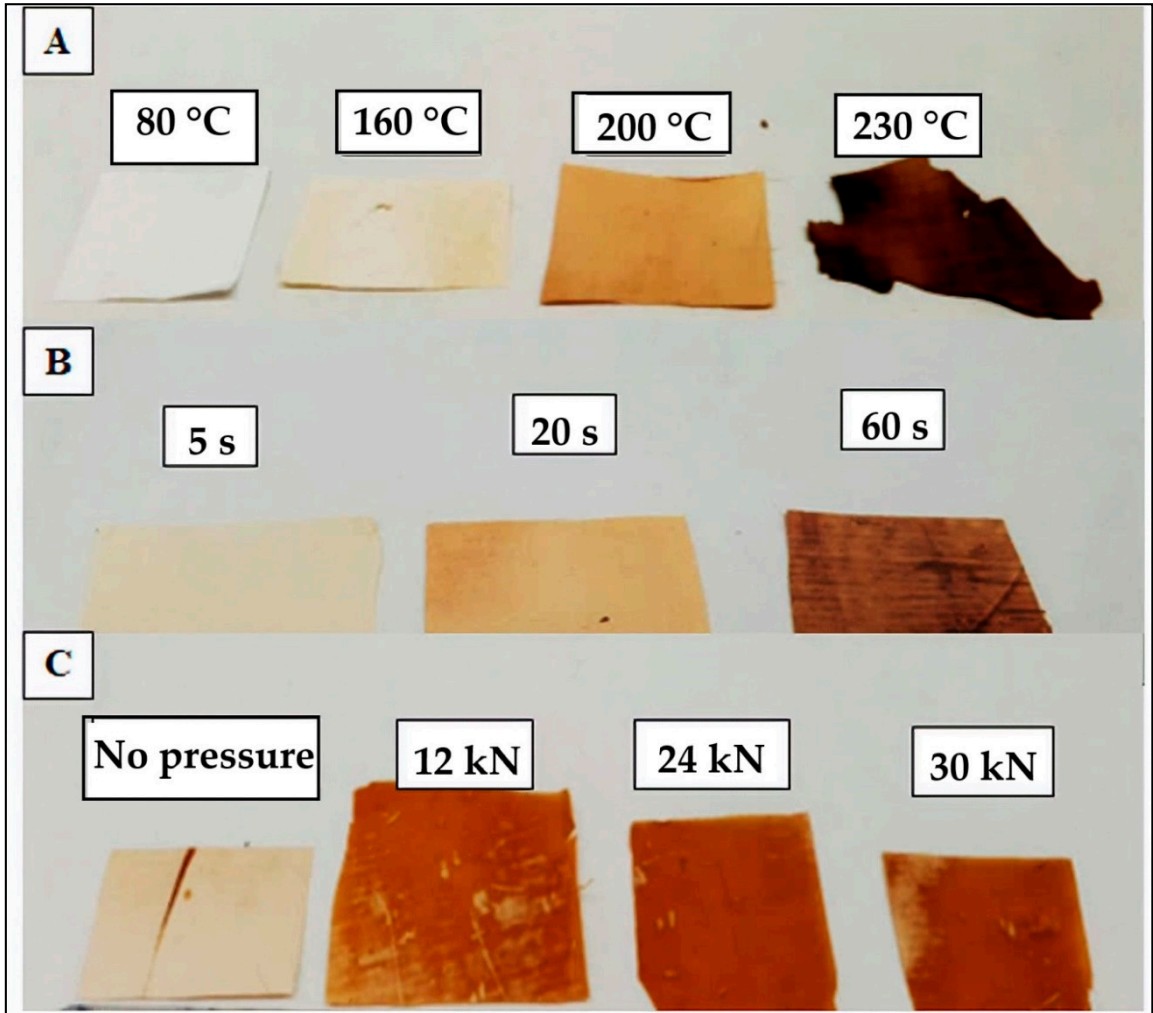

**Figure 4.** Effect of the post-treatment conditions on the pectin-based materials obtained from S3. (**A**) Different temperatures were applied for 1 min without pressure. (**B**) Different times were applied at 190 °C without pressure. (**C**) Different pressures were applied at 165 °C for 60 s.

The above-reported morphological change was further confirmed in Figure 5 by comparison of the SEM images of the cross-sections of the electrospun pectin-based mats from S3 prior to annealing, shown in Figure 5a, and post-processed at 220 °C for 20 s, shown in Figure 5b. This phenomenon can be explained by the oxidation and degradation of pectin compounds at high temperatures. Nevertheless, interestingly, it can be observed that a compact packing rearrangement of the electrospun fibers was observed by a process of fibers coalescence. This phenomenon successfully resulted in the formation of a continuous film. However, some voids were also formed that could result from the evaporation of gases during thermal decomposition.

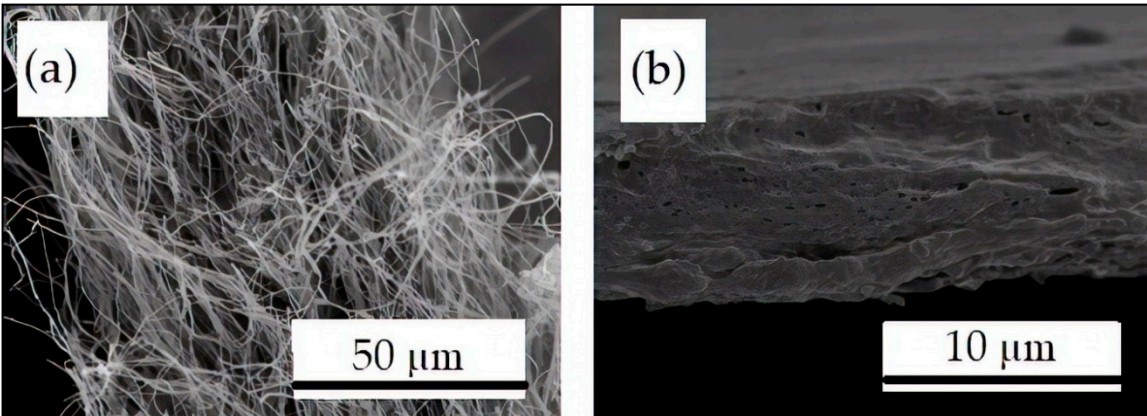

**Figure 5.** Scanning electron microscopy (SEM) images of the fracture surface of the electrospun pectin-based fibers from S3: (**a**) Without any post-treatment and (**b**) Processed at 220 °C for 20 s with a pressure of 24 kN.

Based on the results shown above, the effect on the film-forming process of the two plasticizers, that is, glycerol and $PEG_{900}$, was analyzed. The SEM images of the pectin-based films obtained from the fibers produced with the solutions containing the plasticizers are gathered in Figure 6. The electrospinning of the $PEO_{2000}$-containing pectin solutions with 20–30 wt% glycerol, that is, S5, S6, and S7, resulted in electrospun mats that, after annealing at 150 °C, produced softer and more flexible films. A similar improvement was attained for the electrospun $PEO_{2000}$-containing pectin mats with 20–30 wt% $PEG_{900}$, labelled as S8, S9, and S10, after annealing at 155 °C, though the films were less homogenous. As mentioned earlier, the role of plasticizers in pectin is based on increasing the chain mobility and free volume by creating H-bond interactions with the biopolymer chains [21]. The use of plasticizers thus successfully opened up the post-processability of the electrospun mats by annealing. However, temperatures around 160 °C were still needed, which are certainly close to the $T_{deg}$ values previously measured for pectin and are also responsible for darkening the film samples. Also, for all the formulations, the annealed mats still comprised some porosity, as one can observe in the SEM images gathered in the left column of Figure 6.

To increase the homogeneity of the resultant pectin materials, the electrospun pectin-based fibers were washed with dichloromethane prior to annealing. As seen in the SEM images shown in the middle column of previous Figure 6, the washed fibers partially coalesced, which could potentially enable to reduce the energy requirement for annealing. As mentioned earlier, dichloromethane is a solvent that does not dissolve pectin, but it dissolves glycerol whereas $PEG_{900}$ and $PEO_{2000}$ are slightly soluble. When the pectin fiber mats containing glycerol were immersed in dichloromethane, they did not lose their integrity but, due to the removal of glycerol, the electrospun fibers partially coalesced. Thereafter, annealing of the washed fibers was successful when applied at 150 °C and 140 °C for the fibers obtained from the solutions containing glycerol, that is, S5, S6, and S7, and $PEG_{900}$, that is, S8, S9, and S10, respectively. As it can be observed in the right SEM images of Figure 6, the most homogenous film structures were obtained for the pectin fibers mats with 25 wt% and 30 wt% glycerol and 5 wt% $PEO_{2000}$ were annealed. These results were confirmed by the observation of the optical images of the film samples gathered in Figure 7. Due to the low porosity and completely homogenous film structure, the pectin-based film obtained from S7, that is the pectin formulation with 30 wt% glycerol and 5 wt% $PEO_{2000}$, was selected as the most appropriate candidate for multilayer films.

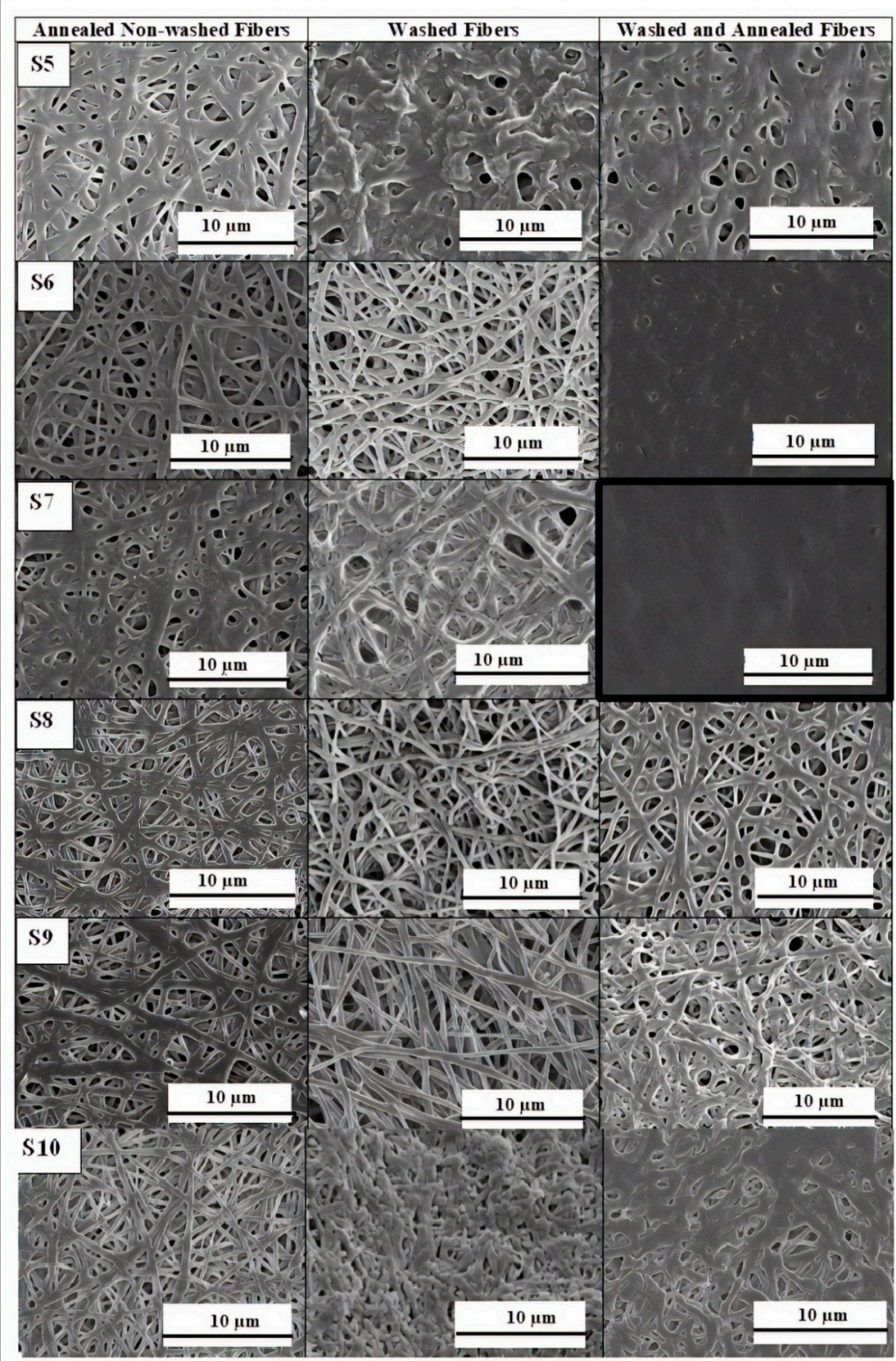

**Figure 6.** Scanning electron microscopy (SEM) images of the electrospun mats after annealing at 150 °C of the pectin-based fibers obtained from S5, S6, and S7 and at 155 °C of the pectin-based fibers obtained from S8, S9, and S10, all processed with a pressure of 24 kN load for 1 min (**left**). Same pectin-based fibers after washing with dichloromethane (**middle**). Washed pectin-based fibers after annealing at 140 °C for the fibers obtained from S5, S6, and S7 and at 150 °C for the fibers obtained from S8, S9, and S10, all processed with a pressure of 12 kN load for 1 minute (**right**). Scale markers of 10 μm in all cases.

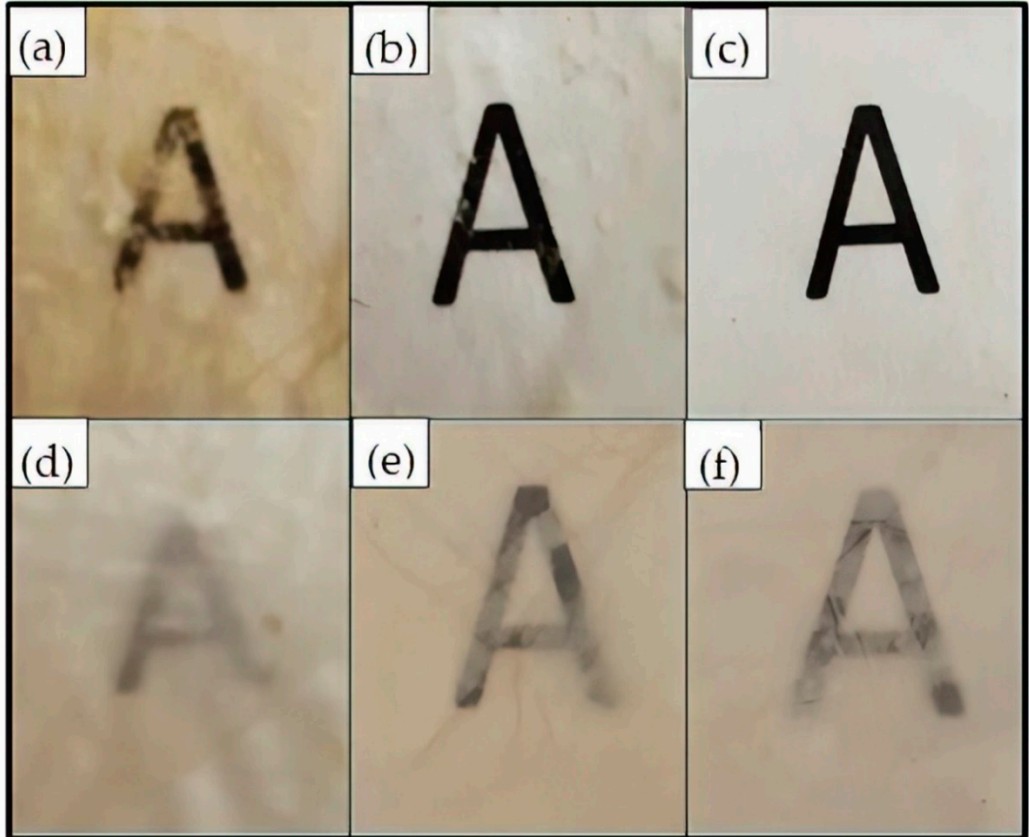

**Figure 7.** Optical images of the electrospun pectin films obtained from the washed fibers of: (**a**) S5; (**b**) S6; (**c**) S7; (**d**) S8; (**e**) S9; (**f**) S10.

### 3.4. Chemical Characterization of Pectin-Based Electrospun Film

ATR-FTIR was carried out on the selected film, obtained from the pectin-based fibers of S7, to ascertain the effect of $PEO_{2000}$ and glycerol on pectin and the thermal post-treatment. The FTIR spectra of the pure components, that is, the pectin powder, glycerol, and $PEO_{2000}$, were also collected. Figure 8 gathers the FTIR spectra of these components and of the pectin-based fibers and film. Table 4 summarizes the band attribution of each peak observed in the FTIR spectra of the pectin materials. One of the characteristic peaks of pectin was seen at 1741 $cm^{-1}$, which refers to the C–H stretching of carboxyl groups (COOH). Also, the two characteristic bands at 1672 $cm^{-1}$ (amide I) and 1595 $cm^{-1}$ (amide II) relate to the presence of amide groups [75,76] of the amidated pectin. The peaks centered at 1132 $cm^{-1}$ and 1070 $cm^{-1}$ may be ascribed to the contribution of the C–C and C–O bonds in secondary alcohol groups of –CH–OH. For glycerol, five characteristic bands at the wavenumbers in the 1150–850 $cm^{-1}$ range arise from the vibrations of C–C and C–O linkages [77]. The FTIR spectrum of $PEO_{2000}$ exhibited $CH_2$ scissoring at 1465 $cm^{-1}$, $CH_2$ wagging at 1357 $cm^{-1}$ and 1340 $cm^{-1}$, $CH_2$ bending at 1280 $cm^{-1}$ and 1240 $cm^{-1}$, C–O–C stretching at 1093 $cm^{-1}$ and 1145 $cm^{-1}$, and $CH_2$ rocking at 962 $cm^{-1}$ and 847 $cm^{-1}$ [78]. The shift in the wavenumber values related to the—CH bending from 1423 $cm^{-1}$, for the pectin powder, to 1411 $cm^{-1}$, for the pectin fibers and film, could be related to a decrease in the interaction between pectin molecules that can be ascribed to the above-described plasticizing effect of glycerol and also to the interaction with $PEO_{2000}$. No changes were observed in the FTIR spectrum of the pectin-based material from fibers to film after the annealing, which proves the absence of chemical reactions and degradation during the applied thermal post-treatment.

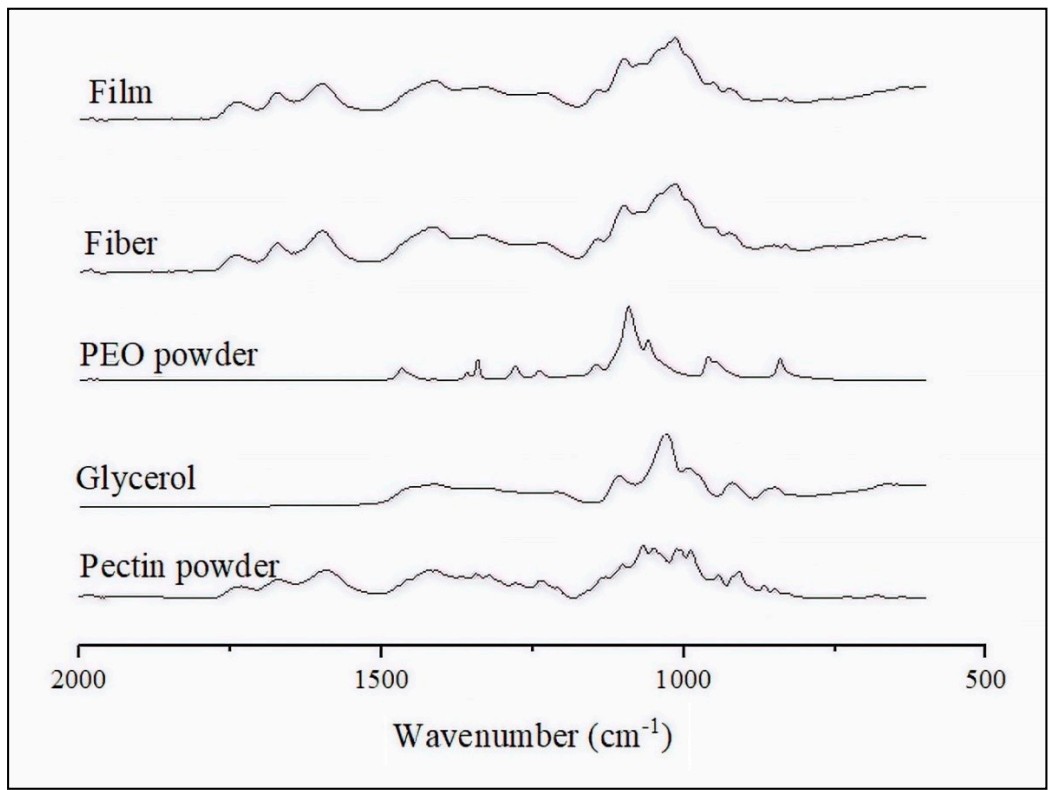

**Figure 8.** Fourier transform infrared spectroscopy (FTIR) spectra, from bottom to top, of pectin powder, glycerol, and polyethylene oxide 2000 (PEO$_{2000}$), and electrospun fibers and film obtained from S7.

**Table 4.** Band attribution of the main peaks observed in the Fourier transform infrared spectroscopy (FTIR) spectra of the pectin-based materials.

| Wavenumber (cm$^{-1}$) | Chemical Group |
|---|---|
| 1741 | C–H stretching |
| 1672 | Amide I |
| 1595 | Amide II |
| 1132 | C–C |
| 1070 | C–O |

*3.5. Application of Pectin-Based Electrospun Film in Multilayers*

The electrospun pectin film was integrated as an inner layer into two electrospun PHBV films, which were prepared as described previously by Cherpinski et al. [51]. The objective of this multilayer was to apply the here-prepared pectin film, which was prepared by the annealing of fibers from S7 solution, as a barrier interlayer that was protected from humidity by two external electrospun layers of PHBV, a bio-based and biodegradable hydrophobic polyester. The whole multilayer structure was formed by electrospinning and subsequent annealing in order to achieve a high adhesion between the layers. A cast film of pectin was also applied in the same conditions for comparison purposes. Figure 9 shows the visual aspect of the electrospun multilayer films to evaluate their contact transparency. The effect of the inner pectin-based interlayer on the color coordinates L*a*b* and ΔE* of the electrospun PHBV films are shown in Table 5. One can observe that all the here-prepared multilayer films presented a good contact transparency. However, a slightly yellowish was developed when the pectin layer was incorporated, particularly for the material obtained by electrospinning and annealing. In particular, the b* values of the PHBV/PHBV multilayer increased from −0.36 to 4.95 and 0.83 for the PHBV/solvent-casted pectin/PHBV and PHBV/electrospun pectin/PHBV multilayer films, respectively. Furthermore, in the case of the PHBV/electrospun pectin/PHBV multilayer, the slight increase in the

a* value suggest that the films became brown. This effect can be related to the original color of the electrospun pectin layers, which was shown in previous Figure 7 and their color values are gathered in Table 5. Therefore, both pectin interlayers generated multilayer films in which an observer can notice different colors ($\Delta E^*_{ab} \geq 5$). In particular, the $\Delta E^*$ values for the PHBV/solvent-casted and PHBV/electrospun pectin/PHBV multilayer films were 9.60 and 11.48, respectively. Additionally, it is worthy to mention that the neat PHBV/PHBV films showed lower transparency than the PHBV monolayers films that were reported in our previous research study [56]. This can be related to possible air entrapment between both PHBV layers during annealing.

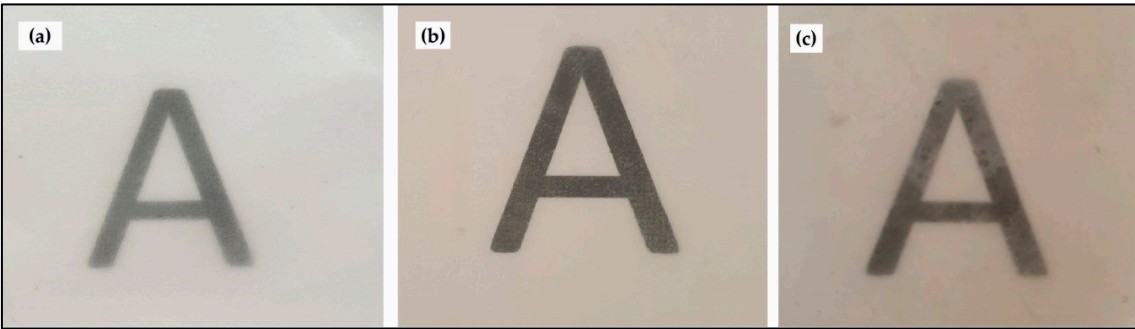

**Figure 9.** Visual aspect of the films based on pectin and poly(3-hydroxybutyrate-*co*-3-hydroxyvalerate) (PHBV): (**a**) PHBV/PHBV multilayer; (**b**) PHBV/solvent-casted pectin/PHBV; (**c**) PHBV/electrospun pectin/PHBV.

**Table 5.** Color coordinates L*, a*, b* and color difference ($\Delta E^*$) of the films based on pectin and poly(3-hydroxybutyrate-*co*-3-hydroxyvalerate) (PHBV).

| Films | L* | a* | b* | $\Delta E^*$ |
|---|---|---|---|---|
| Solvent-casted pectin | 33.21 ± 0.28 [b] | 0.47 ± 0.05 [b] | 1.28 ± 0.12 [b] | - |
| Electrospun pectin | 34.98 ± 0.64 [a] | −0.19 ± 0.03 [a] | 1.53 ± 0.16 [a] | - |
| PHBV/PHBV | 58.26 ± 1.20 [c] | −0.70 ± 0.02 [c] | −0.36 ± 0.05 [c] | - |
| PHBV/solvent-casted pectin/PHBV | 46.82 ± 0.54 [e] | −0.73 ± 0.04 [c] | 0.83 ± 0.09 [e] | 9.60 ± 0.25 [b] |
| PHBV/electrospun pectin/PHBV | 50.58 ± 0.48 [d] | −0.27 ± 0.08 [d] | 4.95 ± 1.17 [d] | 11.48 ± 0.44 [a] |

**a***: red/green coordinates (+a red, −a green), **b***: yellow/blue coordinates (+b yellow, −b blue), **L***: Luminosity (+L luminous, −L dark). [a–e] Different letters in the same column indicate a significant difference ($p < 0.05$).

The cross-sectional SEM images of the obtained multilayer structures are displayed in Figure 10. One can observe in Figure 10a that the bilayer control of PHBV/PHBV formed a continuous structure in which both layers could not be discerned. In relation to the PHBV/pectin/PHBV multilayers, it can be seen that the structure obtained from the electrospun-based pectin film, shown in Figure 10b, presented higher adhesion between layers. However, the multilayer structure based on the cast film of pectin, seen in Figure 10c, easily delaminated during the preparation and observation by SEM. Therefore, multilayer assemblies with a high interlayer adhesion were successfully attained by combining electrospinning and annealing treatments. This effect has been recently attributed to the high aspect ratio of the electrospun fibers, which coalesce during annealing and thus highly adhere to the material substrate [53].

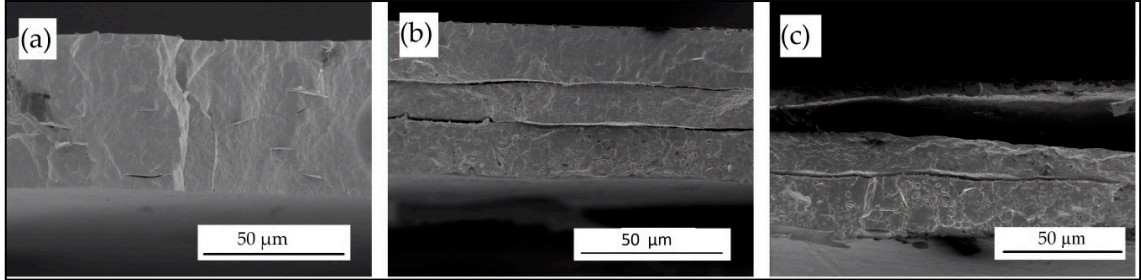

**Figure 10.** Scanning electron microscopy (SEM) images of cross-sections of the multilayer films based on pectin and poly(3-hydroxybutyrate-*co*-3-hydroxyvalerate) (PHBV): (**a**) PHBV/PHBV; (**b**) PHBV/electrospun pectin/PHBV; (**c**) PHBV/solvent-casted pectin/PHBV. Scale markers of 50 μm.

Finally, the permeance of the multilayer films of PHBV/PHBV, PHBV/electrospun pectin/PHBV, and PHBV/solvent-casted pectin/PHBV multilayer films to water and limonene vapors was measured. Permeance is the permeability expression with elimination of the thickness factor used to ascertain the barrier of multilayer structures. On account of comparing the permeance values of multilayer films, thicknesses of the inner and outer layers remained in the same range. Table 6 shows the water and limonene permeance values and the thickness of each layer and the whole structure. In terms of water vapor, the PHBV/PHBV and PHBV/solvent-casted pectin/PHBV multilayer films exhibited somewhat higher permeance values, that is, $5.00 \pm 0.83 \times 10^{-10}$ kg·m$^{-2}$·Pa$^{-1}$·s$^{-1}$ and $3.94 \pm 0.56 \times 10^{-10}$ kg·m$^{-2}$·Pa$^{-1}$·s$^{-1}$, respectively, than the PHBV/electrospun pectin/PHBV multilayers, that is, $1.75 \pm 0.14 \times 10^{-10}$ kg·m$^{-2}$·Pa$^{-1}$·s$^{-1}$. As a result, the incorporation of the cast but more in particular of the electrospun pectin interlayer into PHBV provided enhanced barrier properties against water vapor, even though pectin has intrinsically a hydrophilic character. The reason why the cast film did not have comparable barrier performance to the electrospun film may be ascribed to the lower adhesion at the interphase of the cast material and also to the fact that the casting usually results in less dense film materials due to the slow forming process.

**Table 6.** Water vapor and limonene permeance of the multilayer films based on pectin and poly(3-hydroxybutyrate-*co*-3-hydroxyvalerate) (PHBV).

| Multilayer Structure | Water Vapor Permeance $\times 10^{10}$ (kg·m$^{-2}$·Pa$^{-1}$·s$^{-1}$) | Limonene Permeance $\times 10^{10}$ (kg·m$^{-2}$·Pa$^{-1}$·s$^{-1}$) | Thickness (μm) | | |
|---|---|---|---|---|---|
| | | | PHBV Layers | Pectin Layer | Total |
| PHBV/PHBV | $5.00 \pm 0.83$ [a] | $3.81 \pm 0.47$ [a] | $72 \pm 9$ [a] | - | $72 \pm 9$ [a] |
| PHBV/electrospun pectin/PHBV | $1.75 \pm 0.14$ [b] | $0.22 \pm 0.11$ [b] | $73 \pm 7$ [a] | $25 \pm 5$ [a] | $98 \pm 8$ [b] |
| PHBV/solvent-casted pectin/PHBV | $3.94 \pm 0.56$ [c] | $0.22 \pm 0.08$ [b] | $70 \pm 5$ [a] | $24 \pm 5$ [a] | $95 \pm 5$ [b] |

[a–f] Different letters in the same column indicate a significant difference ($p < 0.05$).

Limonene permeance analysis was also carried out since this organic vapor is often used as an standard for aroma barrier and is also used as an indication for gas barrier properties. From the results, one can observe that the multilayers containing the pectin-based interlayers significantly reduced the limonene permeance, having both the electrospun and solution-casted pectin films the same performance. Limonene permeability is strongly governed by solubility in PHA. Indeed, solubility of limonene in PHBV is relatively high and 100-μm PHBV films can uptake up to 12.7 wt% of limonene [79]. Therefore, it can be considered that the presence of the high barrier to organic vapors, pectin interlayer blocked the passage of the aroma molecules due to its inherent low solubility to limonene. The developed multilayers containing the pectin-based interlayers showed lower permeance than monolayers of polylactide (PLA) electrospun film, that is, $2.62 \pm 1.54 \times 10^{-10}$ kg·m$^{-2}$·Pa$^{-1}$·s$^{-1}$, and PET electrospun film, that is, $0.64 \pm 0.11 \times 10^{-10}$ kg·m$^{-2}$·Pa$^{-1}$·s$^{-1}$ [80].

## 4. Conclusions

Pectin-based electrospun films were successfully produced by electrospinning followed by annealing process. Among the different formulations tested, the water solution of 6.5 wt% pectin, 3.0 wt% glycerol, and 0.5 wt% $PEO_{2000}$ resulted in the production of an electrospun mat composed of defect-free ultrathin fibers that, after annealing, produced the most homogenous and transparent film. Washing fibers with dichloromethane had also a positive effect on the pectin fibers coalescence. The annealing conditions were found optimal at 140 °C and 12 kN for 1 min. The resultant electrospun pectin-based films were finally incorporated as interlayers between two external layers of electrospun PHBV to produce multilayer structures with high barrier properties. The electrospun pectin interlayer successfully decreased the water and limonene vapors barrier values of PHBV and it also showed higher barrier performance when compared with an equivalent multilayer based on a solution-casted pectin interlayer. In this context, the produced electrospun pectin-based films can be considered as sustainable materials to be used for packaging applications of aromatic products.

**Author Contributions:** Conceptualization was carried out by S.A., J.M.L., and S.T.-G.; methodology by B.A.B.; validation and formal analysis, B.A.B. and S.T.-G.; investigation and data curation, B.A.B.; writing—original draft preparation, B.A.B.; writing—review and editing, S.T.-G.; supervision and project administration, S.A., S.T.-G., and J.M.L.

**Funding:** This study was supported by the Turkish Scientific and Technological Research Council (TUBITAK) 2214-A International Research Fellowship Programme for PhD Students and by the Spanish Ministry of Science, Innovation, and Universities (MICIU) project numbers AGL2015-63855-C2-1-R.

**Acknowledgments:** S.T.-G. is a recipient of a Juan de la Cierva—Incorporación contract (IJCI-2016-29675) from MICIU.

**Conflicts of Interest:** The authors have declared no conflict of interest.

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
