# Peer review of "Preparation and Characterization of Electrospun Pectin-Based Films and Their Application in Sustainable Aroma Barrier Multilayer Packaging"

_applsci, doi:10.3390/app9235136_

Round 1

Reviewer 1 Report

The manuscript deals with the preparation and characterization of electrospun pectin-based films and their application in sustainable aroma barrier multilayer packaging.

The English language must be revised.

Abstract

Materials and methods description??This section is confusing and must be revised.

Materials and methods

Color of each film??opacity??solubility in water??elongation at break??tensile strength??

Table 1- Please align text vertically.

Results and discussion

Table 2- Please replace “cp” by “cP”.

Figure 4- Please add another picture. Please separate values from units “80 ºC” and remove green line in “12 kN”, “24 kN” and “30 kN”.

Line 478- “Table 5 shows the water and limonene permeance values and the thickness of each layer and the whole structure.”???Table 5 is missing in the manuscript.

Author Response

The English language must be revised.

The English language was revised and improved along the manuscript.

Abstract

Materials and methods description??This section is confusing and must be revised.

This section is organized based first on the raw materials, then the processing technologies and finally the characterization of the properties of the resultant properties.

Materials and methods

Color of each film??opacity??solubility in water??elongation at break??tensile strength??

Color and contact transparency of pectin films are illustrated on Figure 7 and Figure 9. Also, L, a, b and AE values were measured for the multilayer systems and also for the two selected pectin interlayers and the results are gathered in Table 5. Pectin was already soluble in water since the polymer was not cross-linked or chemically modified by any means. For this reason, the film was applied as an inner layer between two PHBV external layers and the protected to external humidity. Tensile tests were tried on the electrospun films but they could not be properly conducted due to the brittleness of the pectin films. Indeed, the use of the monolayer in packaging applications was technically not feasible. The mechanical properties of the electrospun PHBV films have been reported elsewhere (please see for instance references 53-55). 

Table 1- Please align text vertically.

The text was aligned.

Results and discussion

Table 2- Please replace “cp” by “cP”.

The units symbol were corrected.

Figure 4- Please add another picture. Please separate values from units “80 ºC” and remove green line in “12 kN”, “24 kN” and “30 kN”.

The Figure was amended.

Line 478- “Table 5 shows the water and limonene permeance values and the thickness of each layer and the whole structure.”???Table 5 is missing in the manuscript.

This table was not included in the manuscript by mistake. Thank you for letting us know. You can find it now as Table 6.

Reviewer 2 Report

General comments

The manuscript deals with the preparation and characterization of pectin-based films obtained by electrospinning with the final aim of using the films as aroma barrier multilayer packaging.

The Authors have to better explain their results, trying to give a chemical explanation to the possible interaction between pectin, PEG and plasticizers.

Detailed comments are reported here below.

Detailed comments

line 48: give examples for “conventional ones”

line 51: report some examples of the “different kinds of plasticizers”

lines 53-54: does it mean that you can use pectin without any kind of crosslinking? in that case, pectin cannot have any cohesion, hence any mechanical properties. Please, take into account this comment and revise this part of the Introduction section

line 60: give examples for “articles”

line 64: explain the reason for encapsulation for the final application of your work

lines 84-85: clarify the substrate of the PHB and PHBV coating

lines 108-109: better explain. Did you use two different pectin in your work? Give a reason fro that

lines 142-143: you have to report (in case as SI material) the optimization process that was carried on to get the optimal electrospinning process parameters

line 146: it is not clear if you previously optimized the electrospinning process for PHBV or not. If you optimized the electrospinning parameters in this work, report the optimization process as SI material

line 148: specify the reason for a washing procedure of the electrospun mats. Did you not crosslinked pectin by using divalent ions (e.g., Ca2+)?

line 150: you have to evidence that pectin is not soluble in dichloromethane

line 166: the annealing conditions here reported have been previously optimized and published. But, you have to give a reason for using these parameters without considering the presence of pectin-based layer. Please add a comment in the main text that takes this into account

line 182: you have to report the samples you investigated by DSC and TGA. In addition, you have to report the parameters you considered as output of the thermal characterization you performed (either DSC and TGA)

line 195: you have to report the samples you investigated by FTIR and the information you got from the IR spectra

line 200: you have to report the samples you investigated in the permeability test

line 208: you have to report the time-points you considered (the same for limonene vapor permeability test)

line 226: comment and discussion on the fiber dimension are required as well as the presence of statistical differences among S1 and S10 and all the possible comparisons you can do

lines 227-230: report an image for the electrospinning performed on pristine pectin

line 233: give a reason for the behavior you observed. Which could be the influence of PEO on the pectin-based mat performances?

line 237: describe how you evaluated the “poor integrity”

lines 243-244: “relatively low diameter”, you have to report if you have a requirement for the fiber diameter for your final application

Table 2: you have to clarify the meaning of the superscript letters

lines 257-276: 1) highlight in the main text the significant differences you detected comparing your solution compositions; 2) you have to try to correlate the values of viscosity, surface tension and conductivity to check any possible correlation among them (lines 275-276)

line 277: you have to explain the possible chemical interaction between pectin and plasticizers that caused different properties and different fiber diameter

lines 307-310: you have to give a chemical possible explanation for the shift you detected in the exothermic peak of pectin thermal degradation

line 318: for TGA data you have to add a statistical analysis that evidence possible significant differences among the different compositions you investigated

line 325: you have to explain to what the “second weight loss” is related

lines 333-334: better explain what happens for the LM pectin you used

lines 344-345: give a reason for this behavior

Table 3: add any statistical difference in the thermal parameters you investigated

line 363: you have to justify and explain the rationale for processing pectin by electrospinning if you want to obtain “homogeneous and continuous films”. It should be easier to prepare directly compact films

line 370: “low degradation temperature of pectin”, specify to which temperature you are referring

line 379: “became too brittle to be applied as packaging materials”, you have to support this conclusion by quantitative data

lines 384-391: what about the other possible treated pectin-based samples?

line 400: “softer and more flexible”, you have to perform mechanical tests to assert that

line 404: you have to validate this conclusion by FTIR analysis

line 434: 1) report the band attribution in a table; 2) evidence the main peaks on the spectra reported in Figure 8; 3) try to support your findings by literature

line 460: “electrospinning and annealing”, better explain: you performed a second annealing on the multilayer film?

line 466: you cannot assert the “higher adhesion between layers” by SEM observation. Mechanical tests are required to evidence the strength between the different layers in PHBV/pectin/PHBV film

lines 468-470: you have to give a reason for this behavior

lines 475-499: Table 5 is not present in the manuscript (as well as Table 4 …). Hence, it is not possible to understand your findings. Moreover, you have to compare your data to the traditional barrier films used in packaging. You have to clarify if the thickness of the here obtained PHBV/pectin/PHBV film is in the adequate range (compared to the traditional films). In addition, for packaging applications, mechanical properties as well as degradation rate have to be considered. These data are not present in the manuscript and are not considered and discussed.

Author Response

General comments

The manuscript deals with the preparation and characterization of pectin-based films obtained by electrospinning with the final aim of using the films as aroma barrier multilayer packaging.

The Authors have to better explain their results, trying to give a chemical explanation to the possible interaction between pectin, PEG and plasticizers.

Some discussion was added in the manuscript about the potential substitution of the intermolecular H-bonds of pectin by plasticizer-pectin H-bonds and covalent esters.

Detailed comments are reported here below.

Detailed comments

Line 48: give examples for “conventional ones”

Some examples of conventional polymers were added. Please see page 2, lines 49-51.

Line 51: report some examples of the “different kinds of plasticizers”

Some examples of plasticizers were added. Please see page 2, lines 53-57.

Lines 53-54: does it mean that you can use pectin without any kind of crosslinking? in that case, pectin cannot have any cohesion, hence any mechanical properties. Please, take into account this comment and revise this part of the Introduction section

Cross-linking is habitually performed to improve the water resistance and mechanical cohesion of pectin in most applications. However, in this study it was not needed since the electrospinning process makes use of water solutions of pectin and the resultant mats were applied as an interlayer between two hydrophich biopolymer films.

Line 60: give examples for “articles”

The term “Articles” was replaced by “films” to avoid confusion.

Line 64: explain the reason for encapsulation for the final application of your work

This sentence has been removed to avoid misunderstanding.

Lines 84-85: clarify the substrate of the PHB and PHBV coating

The substrate was defined in lines 86-88, page 2.

Lines 108-109: better explain. Did you use two different pectin in your work? Give a reason fro that.

No, A single type of pectin was used but we added different contents of PEO and tested 2 different plasticizers (glycerol or PEG900). This is explained in page 3, lines 103-105, at the end of the Introduction.

Lines 142-143: you have to report (in case as SI material) the optimization process that was carried on to get the optimal electrospinning process parameters

The optimization process during electrospinning was explained in page 4, lines 151-156 and 158.

Line 146: it is not clear if you previously optimized the electrospinning process for PHBV or not. If you optimized the electrospinning parameters in this work, report the optimization process as SI material

It was previously optimized during a previous study, please see reference 54.\

Line 148: specify the reason for a washing procedure of the electrospun mats. Did you not crosslinked pectin by using divalent ions (e.g., Ca2+)?

The electrospun mats were washed with dichloromethane in order to facilitate coalescence of the electrospun fibers. Indeed, cross-linking could prevent annealing and would produced an opposite effect. This was explained in page 4, lines 162-163.

Line 150: you have to evidence that pectin is not soluble in dichloromethane

A reference to support this statement was added. Please reference 39.

Line 166: the annealing conditions here reported have been previously optimized and published. But, you have to give a reason for using these parameters without considering the presence of pectin-based layer. Please add a comment in the main text that takes this into account.

An explanation for the selection of these annealing conditions was added. Please see page 5, lines 181-182.

Line 182: you have to report the samples you investigated by DSC and TGA. In addition, you have to report the parameters you considered as output of the thermal characterization you performed (either DSC and TGA)

This was indicated in page 5, lines 198-200 and line 207.

Line 195: you have to report the samples you investigated by FTIR and the information you got from the IR spectra

This was also indicated in pages 5 and 6, lines 213-214 and 217-218.

Line 200: you have to report the samples you investigated in the permeability test.

This was added in page 6, lines 236 and 248.

Line 208: you have to report the time-points you considered (the same for limonene vapor permeability test)

This was added in page 6, lines 242.

Line 226: comment and discussion on the fiber dimension are required as well as the presence of statistical differences among S1 and S10 and all the possible comparisons you can do

More discussion was added in section 3.1.

Lines 227-230: report an image for the electrospinning performed on pristine pectin

Neat pectin was not processable by electrospinning and large droplets were formed that were not observable by SEM.

Line 233: give a reason for the behavior you observed. Which could be the influence of PEO on the pectin-based mat performances?

The effect of PEO in the electrospinnability of pectin was described in page 7, lines 268-269. This was further discussed based on the solution properties.

Line 237: describe how you evaluated the “poor integrity”

This was described in page 7, lines 274-275.

Lines 243-244: “relatively low diameter”, you have to report if you have a requirement for the fiber diameter for your final application

There was no certain requirement for the fiber diameter for our final application but it was necessary to obtain continuous and free-bead fibers in order to produce films. This was indicated in page 7, lines 283-284.

Table 2: you have to clarify the meaning of the superscript letters

This was better explained in the table caption.

Lines 257-276: 1) highlight in the main text the significant differences you detected comparing your solution compositions; 2) you have to try to correlate the values of viscosity, surface tension and conductivity to check any possible correlation among them (lines 275-276).

The observable differences on solution properties (viscosity, surface tension and conductivity) with the change in solution compositions were discussed in page 9.

Line 277: you have to explain the possible chemical interaction between pectin and plasticizers that caused different properties and different fiber diameter

Some discussion about the possible chemical interaction between pectin and plasticizers was added in page 10, lines 321-331 and 336-338.

Lines 307-310: you have to give a chemical possible explanation for the shift you detected in the exothermic peak of pectin thermal degradation.

This thermal degradation delay was explained in page 10, lines 361-363.

Line 318: for TGA data you have to add a statistical analysis that evidence possible significant differences among the different compositions you investigated

Statistical analysis was applied on the data reported in Table 3.

Line 325: you have to explain to what the “second weight loss” is related

This was described in page 11, lines 386-388.

Lines 333-334: better explain what happens for the LM pectin you used

It was explained in page 11, lines 390-393.

Lines 344-345: give a reason for this behavior.

New information was added in pages 12, lines 404-407.

Table 3: add any statistical difference in the thermal parameters you investigated.

Statistical analysis was added.

Line 363: you have to justify and explain the rationale for processing pectin by electrospinning if you want to obtain “homogeneous and continuous films”. It should be easier to prepare directly compact films.

Neat pectin cannot be processed by conventional melt processing technologies and the films are only obtained the solvent casting method. This was already indicated in the Introduction, page 2, lines 62-64.

Line 370: “low degradation temperature of pectin”, specify to which temperature you are referring

These temperatures are shown in Table 3.

Line 379: “became too brittle to be applied as packaging materials”, you have to support this conclusion by quantitative data

The pectin layers were too brittle for mechanical testing. So, the mechanical analysis could not be done. Also our intention was to use pectin film as an internal monolayer not as a single film.

Lines 384-391: what about the other possible treated pectin-based samples?

We ruled out of the study some of the pectin fibers by the reasons that were explained in the first paragraph of section 3.1.

Line 400: “softer and more flexible”, you have to perform mechanical tests to assert that.

The pectin layers were too brittle for the mechanical testing and tensile tests were not feasible in universal testing machines.

Line 404: you have to validate this conclusion by FTIR analysis

This is explained in page 17, lines 516-519.

Line 434: 1) report the band attribution in a table; 2) evidence the main peaks on the spectra reported in Figure 8; 3) try to support your findings by literature

Please see new Table 4 and section 3.3.

Line 460: “electrospinning and annealing”, better explain: you performed a second annealing on the multilayer film?

A detailed explanation of the multilayers preparation is included in the second paragraph of section 2.6.

Line 466: you cannot assert the “higher adhesion between layers” by SEM observation. Mechanical tests are required to evidence the strength between the different layers in PHBV/pectin/PHBV film

This work has been rather focused on the development of pectin-based layers by electrospinning and its use in barrier packaging. Mechanical tests for electrospun PHB and PHBV films have widely performed in previous works (see for instance references 53, 54 or 55), which are the structural layers that control the mechanical performance of the films. Moreover, tensile tests are habitually not useful to determine adhesion since this is based on the application of uniaxial forces.

Lines 468-470: you have to give a reason for this behavior.

This was explained in lines 57-5731, page 19.

Lines 475-499: Table 5 is not present in the manuscript (as well as Table 4 …). Hence, it is not possible to understand your findings. Moreover, you have to compare your data to the traditional barrier films used in packaging. You have to clarify if the thickness of the here obtained PHBV/pectin/PHBV film is in the adequate range (compared to the traditional films). In addition, for packaging applications, mechanical properties as well as degradation rate have to be considered. These data are not present in the manuscript and are not considered and discussed.

The tables was added. Please see new Table 6. Since this study deals with permeance of multilayer films these values cannot be directly compared with permeability values of conventional polymers. However, we have discussed the improvement in permeance in relation to the PHBV/PHBV. The latter value can be related to the permeability with other monolayers (taken into account its thickness value) and this has been already done in our previous works and it was also compared, as an example, with that of PET (please see page 20, lines 599-601).

Reviewer 3 Report

Nice study, I recommend it for publication.
Please rewrite the abstract describing briefly the aim of the study, the results achieved and the potential uses or the relevant scientific finding or technological advantage of the study.
I couldn't find the description of Span 20 in 2.1.
If possible, please improve the quality of figures 1-3, 8 (vector image if possible). It would be better to place sample numbers near each curve instead of using a separated legend in figure 1 and 2.

Author Response

Nice study, I recommend it for publication.

Please rewrite the abstract describing briefly the aim of the study, the results achieved and the potential uses or the relevant scientific finding or technological advantage of the study.

The aim of the study and potential uses of the material developed were better described in the section “Featured Application”, which can be seen above the abstract. The abstract now includes only the performed research activities and the relevant results achieved.

I couldn't find the description of Span 20 in 2.1.

A full description of this raw material was added in page 3, lines 123-125.

If possible, please improve the quality of figures 1-3, 8 (vector image if possible). It would be better to place sample numbers near each curve instead of using a separated legend in figure 1 and 2.

The figures were improved according to the resolution required by the journal. 

Round 2

Reviewer 1 Report

Materials and methods

Line 220- Color measurements, illuminant and ºobserver used?????

Results and discussion

Table 2- Please align text vertically.

Table 5- Please organize the order by L*a*b*.

Author Response

Materials and methods

Line 220- Color measurements, illuminant and ºobserver used?????

The type of illuminant and degree observer used was added.

Results and discussion

Table 2- Please align text vertically.

The table format was amended.

Table 5- Please organize the order by L*a*b*.

The table data was reorganized.

Reviewer 2 Report

The Authors have adequately revised the manuscript improving the quality of the work

Author Response

The Authors have adequately revised the manuscript improving the quality of the work.

Thank you very much reviewing the manuscript. Your comments have helped a lot to improve its quality.